# The First *Yarrowia lipolytica* Yeast Models Expressing Hepatitis B Virus X Protein: Changes in Mitochondrial Morphology and Functions

**DOI:** 10.3390/microorganisms10091817

**Published:** 2022-09-10

**Authors:** Khoren K. Epremyan, Tatyana N. Goleva, Anton G. Rogov, Svetlana V. Lavrushkina, Roman A. Zinovkin, Renata A. Zvyagilskaya

**Affiliations:** 1A.N. Bach Institute of Biochemistry, Research Center of Biotechnology of the Russian Academy of Sciences, Leninsky Ave. 33/2, 119071 Moscow, Russia; 2National Research Center “Kurchatov Institute”, Akademika Kurchatova pl. 1, 123182 Moscow, Russia; 3Belozersky Institute of Physico-Chemical Biology, Lomonosov Moscow State University, Leninskye Gory 1/40, 119992 Moscow, Russia; 4Faculty of Bioengineering and Bioinformatics, Lomonosov Moscow State University, Leninskye Gory 1/73, 119234 Moscow, Russia

**Keywords:** hepatocellular carcinoma, hepatitis B virus, HBx, heterologous expression, yeast, *Yarrowia lipolytica*, oxidative stress, mitochondrial dysfunction, mitochondria-targeted antioxidants

## Abstract

Chronic hepatitis B virus infection is the dominant cause of hepatocellular carcinoma, the main cause of cancer death. HBx protein, a multifunctional protein, is essential for pathogenesis development; however, the underlying mechanisms are not fully understood. The complexity of the system itself, and the intricate interplay of many factors make it difficult to advance in understanding the mechanisms underlying these processes. The most obvious solution is to use simpler systems by reducing the number of interacting factors. Yeast cells are particularly suitable for studying the relationships between oxidative stress, mitochondrial dynamics (mitochondrial fusion and fragmentation), and mitochondrial dysfunction involved in HBx-mediated pathogenesis. For the first time, genetically modified yeast, *Y. lipolytica*, was created, expressing the hepatitis B virus core protein HBx, as well as a variant fused with eGFP at the C-end. It was found that cells expressing HBx experienced stronger oxidative stress than the control cells. Oxidative stress was alleviated by preincubation with the mitochondria-targeted antioxidant SkQThy. Consistent with these data, in contrast to the control cells (pZ-0) containing numerous mitochondrial forming a mitochondrial reticulum, in cells expressing HBx protein, mitochondria were fragmented, and preincubation with SkQThy partially restored the mitochondrial reticulum. Expression of HBx had a significant influence on the bioenergetic function of mitochondria, making them loosely coupled with decreased respiratory rate and reduced ATP formation. In sum, the first highly promising yeast model for studying the impact of HBx on bioenergy, redox-state, and dynamics of mitochondria in the cell and cross-talk between these parameters was offered. This fairly simple model can be used as a platform for rapid screening of potential therapeutic agents, mitigating the harmful effects of HBx.

## 1. Introduction

Liver cancer is a prevalent cancer with a rapidly increasing incidence and a major contributor to cancer-related death with poor survival for sufferers [1]. Hepatocellular carcinoma (HCC), an aggressive human malignancy, is the most common form of liver cancer [2,3,4]. According to the latest epidemiological data from the Globocan 2020 report, HCC accounts for 4.7% of all cancers, with more than 900,000 new cases and about 830,000 deaths associated with HCC each year [3]. It is the fifth most prevalent cancer worldwide, and the third major cause of cancer mortality globally [5,6,7,8,9].

The prevalence of the high mortality rate is due to the lack of sensitive and specific biomarkers for early detection [10], limited treatment options for late-stage HCC [10], drug resistance, invasiveness, and metastasis [6,11]. Remarkable differences were found in HCC incidence and mortality in gender, race/ethnicity, age groups, and geographical regions. The highest rates of HCC are typical for East and Southeast Asia and Central and South Africa [12,13,14], but recently there has been a trend towards an increase in the number of cases also in the United States and Western Europe [15].

The precise molecular mechanisms that mediate HCC development are still unclear, but chronic hepatitis B virus (HBV) infection is considered to be the major risk factor for HCC initiation, and most patients suffered from HCC as a consequence of persistent HBV infection [5,16,17]. Despite the development of vaccines and therapeutic approaches against HBV, 2 billion people have been infected with HBV [1]. The propagation of HBV infection depends on environmental factors, lifestyle, earning levels, and education, which determine differences in the geographical distribution of HBV incidence, being about 1% in developed countries and 8% or even more in the rest. HBV can be transmitted sexually and perinatally or by direct exposure to contaminated blood or other body fluids, and it attacks the liver to induce acute and chronic diseases [18]. Differences in HBV genotype and mutations, germ line genetic variation predisposition to the host, acquisition of tumor-specific somatic mutations, and environmental factors contribute to the observed individual variability in the development of HCC [19]. Chronic infection with the HBV, despite the availability of a vaccine, remains quite common [2].

HBV is a member of the Hepadnaviridae family and the *Orthohepadnavirus* genus. The HBV genome, which is highly compact (only 3.2 kilobases in length), is a relaxed circular, partially double-stranded DNA molecule that contains four overlapping open reading frames encoding envelope protein (pre-S1/pre-S2), core protein (pre-C/C), viral polymerase, and X protein (HBx) [20]. Among the proteins produced by HBV, HBx protein is strongly associated with HCC development. Replication of the hepatitis B virus is a complex multistep process. Initially, nucleocapsids enter the hepatocyte nucleus through the NTCP receptor, where the RC DNA is converted into a covalently closed circular DNA (cccDNA). For the initiation and maintenance of virus replication, the presence of the HBx protein, which has transcriptional activation activity, is necessary. Then, a 3.5 kb pregenomic RNA is transcribed from cccDNA and transported to the cytoplasm, where mature capsids are secreted outward, or transferred back to the nucleus, forming a cccDNA pool. Due to this mechanism, some HBV genes can be integrated into the chromosomal DNA of infected hepatocytes [2,17,21,22]. Until now, not all molecular mechanisms of HBV-mediated carcinogenesis have been identified. It is assumed that the immune response is basically associated with chronic inflammation and integration of the viral genome into the hepatocyte genome, and the key role in carcinogenesis is played by the viral regulatory protein HBx [5,23]. 

The 17 kDa non-structural protein HBx, encoded by the X region, is a multifunctional nonspecific transactivator. HBx modulates cytoplasmic signal transduction and directly interacts with nuclear transcription factors, which makes it possible to regulate not only viral but also cellular promoters. The HBx structure is still unknown, but a number of in silico models of HBx have been recently proposed [24,25]. The C-terminal fragment of HBx, being a significant element of a spatial protein structure, starts at the 120 position (tryptophan residue) and is exposed to the cytoplasm. The 113–135 residues, known as BH3-like peptide, are involved in the regulation of HBV replication [25]. The C-terminal fragment plays a crucial role in carcinogenesis [26] and is required for ROS production by hepatocyte mitochondria [27]. Integration of the viral genome is accompanied by truncation of the C-terminal fragment of the HBx protein (ct-HBx), which further accelerates carcinogenesis [28] by sustaining proliferative signaling [29], evading growth suppressors [30], evading immune destruction [31,32], facilitating replicative immortality [33], aiding in tumor-promoting inflammation [34], triggering invasion and metastasis [35,36], prompting angiogenesis [37], and inducing genome instability [38]. The role of HBx in the induction of apoptosis in HCC is contradictory and depends to a large extent on the cellular conditions, components interacting with HBx, and intracellular localization of wild-type HBx (wtHBx) and truncated forms of HBx (trHBx) [39]. 

Although it is widely known that HBx is a multifunctional regulator and an attractive therapeutic target for the treatment of chronic hepatitis B and HCC, the specific molecular mechanisms of HBV-associated HCC, as well as the role of the HBx protein in carcinogenesis, are not well understood and require further research [40,41]. Findings of HBx effects on cell energy metabolism are scarce and contradictory [42].

The complexity of the system itself and the intricate interplay of many factors make it difficult to advance the understanding of the mechanisms underlying these processes. The most obvious solution is to use simpler systems and reduce the number of interacting factors. 

Indeed, the use of relevant cell lines has provided valuable new information and has shed light on the underlying molecular mechanism and novel therapy targets for HBx, HBV, and HCC. We quote only a few of the most striking works. A detailed review of the works deserves a separate publication.

A novel monoclonal antibody was developed that enables a spatiotemporal analysis of HBx in a natural infection system in HBV-infected primary human hepatocytes. Confocal imaging studies with this antibody demonstrated that HBx is expressed shortly after infection and has a short half-life and that it is predominantly located in the nucleus [43,44]; in contrast to the general belief that HBx is mostly cytoplasmic, with a small fraction in the nucleus, the mitochondrion is a major target for HBx in the cytoplasm [43].

Down-regulation of miR-30c may result in the progression of chronic HBV via the promotion of HBV replication and cell proliferation [45]. The ability of the full-length HBx protein and its truncated forms to act on the cell cycle of regulatory proteins is one of the main elements of pathogenesis in the development of HCC [46].

HBx promotes the proliferation, epithelial–mesenchymal transition, invasion, and migration of HCC cells by targeting HMGA2, a potential therapeutic target for HBV-associated HCC [47]; regulates diverse aspects of LONP1 and Parkin, enhancing mitophagy in starvation [48]; promotes HCC metastasis by remodeling the extracellular matrix modification through the HIF-1α/LOX pathway [49]; and alters the expression of long non-coding RNAs to promote the progression of HCC [7].

HBx-induced S100A9 plays a pivotal role in the metastasis of HCC [50]; SHP2 induced by the HBx-NF-κB pathway contributes to fibrosis during human early HCC development [51], and the interaction between centrosomal P4.1-associated protein (CPAP) and HBx provides a microenvironment to facilitate HCC development by enhancing NF-κB activation, inflammatory cytokine production, and cancer malignancies [52].

The dual roles of cellular FLICE inhibitory protein (c-FLIP) in the regulation of HBV replication was found; c-FLIP interacts with HBx and enhances its stability and regulates the expression or stability of hepatocyte nuclear factors, which are essential for transcription of the HBV genome [40].

Thioredoxin-interacting protein (TXNIP), a key mediator of intracellular ROS, may be involved in HBx-mediated metastasis of HBV-associated HCC [53]. The autophagy-related protein 16-1 (ATG16L1) binds to the ATG12-ATG5 conjugate and forms a large protein autophagosome complex involved in HBV-associated HCC [54].

While applying relevant cell lines has greatly contributed to better understanding of some HCC-related processes, they remain complex with a number of drawbacks, including low cell growth rates, dense cell association, and possible impacts of the rich cultivation medium. Yeast models are devoid of these flaws. Yeast cells, the simplest eukaryotic organisms, sharing well-preserved universal molecular and cellular mechanisms regulating signaling pathways [55,56], proteostasis, autophagy, oxidative stress, secretory pathways [57,58], and cell death [58], are particularly suitable for studying the relationships between oxidative stress, mitochondrial dynamics (mitochondrial fusion and fragmentation), and mitochondrial dysfunction. Moreover, due to their ability to grow rapidly on simple accessible growth media of certain compositions, their advanced developed genetic toolbox, and their expanded applications of synthetic biology and metabolic engineering, yeasts have become a valuable eukaryotic model organism for unraveling the complex intracellular mechanisms underlying human biology and pathology [59,60,61]. In addition, yeast cells are naturally devoid of HBx protein, which allows researchers to determine morphological and bioenergy changes in cells under the influence of viral protein HBx separately from other factors (chronic inflammation, immune and cytokine response, etc.) causing oxidative stress as a secondary process.

The non-pathogenic, non-toxic (generally regarded as safe), obligate aerobic ascomycetous yeast *Y. lipolytica* cells, having respiratory metabolism closely resembling that of mammalian cells [62,63,64,65], a versatile substrate utilization profile, rapid growth rate, developed advanced genome editing technologies, and unique physicochemical properties and secretory machinery, contributing to the extraordinary capacity for production and secretion of heterologous proteins (see [66]), is especially appropriate for this kind of research.

The main goal of this study is to investigate the cross-talk between redox and energy status and dynamics of mitochondria in *Yarrowia lipolytica* yeast cells expressing HBx and its variants. 

## 2. Materials and Methods

### 2.1. Chemical Reagents

Bacto agar, Bacto peptone, Bacto yeast extract, Dithiothreitol (DTT), and Tris (ultra-pure) were purchased from Becton, Dickinson and Company (Franklin Lakes, New Jersey, USA); ADP, ampicillin, Anti-Rabbit IgG Peroxidase antibody, antimycin A, ATP, 3-amino-1,2,4-triazole, carbonyl cyanide m-chlorophenylhydrazon(CCCP), P5-di(adenosine-5)pentaphosphate (Ap5A), EDTA, EGTA, fatty acid-free BSA, glucose, glucose-6-phosphate dehydrogenase, LiAc, mannitol, MgCl_2_, NaCl, NADP, (NH_4_)_2_SO_4_, oligomycin, Phenol Red, phosphoenolpyruvate, pyruvate kinase, rotenone, succinic acid, and *tert*-butyl hydroperoxide were from Sigma-Aldrich (St. Louis, MO, USA); Coomassie G-250 and zymolyase were from MP Biomedicals (Santa Ana, California, USA); CaCl_2_, K_2_HPO_4_, KCl, KH_2_PO_4_, NaCl, and safranin O were from Merck (Darmstadt, Germany); 10× DNA Loading Dye, 10x G+ buffer, 10x O+ buffer, 10x R+ buffer, BSA, Dihydroetidium, Sytox Green Dead Cell Stain, DMSO, Gene Jet Gel Extraction Kit, Gene Jet Plasmid Miniprep Kit, Gene Ruler 100 bp+, Gene Ruler 1 kb, Glycogen, Mitotracker Red CmxRos, NotI restriction endonuclease, PageRuler™ Prestained Protein Ladder, Phusion High-Fidelity PCR Kit, PvuII restriction endonuclease, Rapid DNA Ligation Kit, RNAse-A, SuperSignal™ West Dura Extended Duration Substrate, and XhoI restriction endonuclease were from Thermo Fisher Scientific (Waltham, MA, USA); agar, agarose LE2, ethidium bromide, glycerol (ultra-pure) were from Helicon (Moscow, Russian Federation); LB BROTH Miller (Luria–Bertani) and NaOAc were from Amresco (Dallas, Texas, USA); BbsI (BpiI) restriction endonuclease was from New England Biolabs (Ipswich, Massachusetts, USA); sorbitol was from Dia-M (Moscow, Russian Federation); oligonucleotides were from DNA-Synthesis (Moscow, Russian Federation); Anti-GFP antibody was from Evrogen (Moscow, Russian Federation). SkQThy was kindly provided by Dr. Esipov D.S. from A. N. Belozersky Research Institute of Physico-Chemical Biology MSU, Moscow, Russian Federation.

### 2.2. Cell Cultures

*Escherichia coli* strain XL1-Blue (Evrogen, Russian Federation) was used for plasmid propagation. Cells were grown on Petri dishes with sterile LB medium (1% Bacto peptone, 0.5% yeast extract, 1% NaCl) containing 2% Bacto agar and supplemented with 100 μg/mL ampicillin as a selectivity factor overnight at 37 °C.

The *Y. lipolytica* yeast, strain Po1f, was obtained from National Bioresource Center—All-Russian Collection of Industrial Microorganisms (Moscow, Russian Federation). All the strains used in this study are listed in Table 1. *Y. lipolytica* cells were grown in 500 mL Erlenmeyer flasks at 28 °C on a rotary shaker ES-20/60 (Biosan, Rīga, Latvia) at 220 rpm on a semi-synthetic medium [67] containing 1.3% succinate as a carbon and energy source and harvested in the early exponential growth phase (OD = 1).

### 2.3. Plasmid and Yeast Strain Construction

Primer design was based on the nucleotide sequences of the genes encoding HBx and eGFP so that the sequences of the PCR products consisted, respectively, of the full-length HBx nucleotide sequence and the full-length eGFP nucleotide sequence.

The HBx open reading frame was amplified from the pCMV-Sport6-HBx plasmid [68] by using pairs of primers, namely, HBx-BbsI-Fw1/HBx-BbsI-Rev1, for pZ-HBx construction, and HBx-BbsI-Fw1/HBx-BbsI-rev2 for pZ-HBx-eGFP.

The eGFP open reading frame was amplified from pUC:FCP:ShBle:FCP:EGFP (Addgene, Watertown, MA, USA) by using pairs of primers, namely, eGFP-BbsI-Fw1/eGFP-BbsI-Rev1 for pZ-eGFP, and eGFP-BbsI-Fw2/eGFP-BbsI-Rev1 for pZ-HBx-eGFP constructions. 

Primer sequences used in this study are listed in Table 2. The absence of BbsI restriction sites in the target gene sequences was determined, and the lack of autocomplementarity of the ends of the PCR products was estimated using SnapGene software (GSL Biotech LLC, San Diego, CA, USA).

Polymerase chain reactions were carried out using Phusion Hot Start II high-fidelity DNA polymerase. PCR mixtures contained Phusion HF buffer, 200 μM dNTPs, the corresponding primers (final concentration, 0.5 μM), and 10 ng of matrix DNA. PCR fragments were purified from agarose gels using a Gene Jet Gel Extraction Kit.

To create target genetic constructs, the pZ-express++ plasmid with a hybrid hp4d promoter dependent on the growth phase and a ZETA transposon sequence with multiple homology in the *Y. lipolytica* genome was chosen, which ensures a high copy number of the plasmid during recombination and, as a result, a high level of expression of the target protein. The plasmid also has an ampicillin-resistance gene and a prototrophic factor for uracil URA3 from the *Y. lipolytica* genome. Insertion of the PCR products into the pZ-express++ vector was performed by the Golden Gate Cloning method using BbsI type II restriction enzyme and T4 ligase in T4 ligase buffer. The sequences of HBx and HBx-eGFP were inserted into the vector. All plasmid constructs were verified by restriction enzyme mapping and DNA sequencing of inserted fragments. The pZ-express++ plasmid was kindly provided by Dr. Laptev I.A. from Federal Institution “State Research Institute of Genetics and Selection of Industrial Microorganisms of the National Research Center” Kurchatov Institute”, Moscow, Russian Federation.

The transformation of *E. coli* XL1-Blue cells with genetic constructs was carried out by electroporation at 1720 V using a MicroPulser Electroporator (Bio-Rad, USA). The cells carrying plasmids with target genes were allowed to grow on Petri dishes with sterile LB medium containing 2% Bacto agar and supplemented with 100 μg/mL ampicillin overnight at 37 °C. Positive clones carrying correct plasmids were isolated and cultivated in liquid sterile LB medium supplemented with 100 μg/mL ampicillin overnight at 37 °C. 

Plasmids were isolated using a GeneJet Plasmid Miniprep Kit, and the DNA concentrations were determined spectrophotometrically with a NanoDrop 1000 Spectrophotometer (Thermo Fisher Scientific, USA) at a wavelength of 260 nm. Plasmids were linearized with NotI endonuclease and used for transfection of *Y. lipolytica* Po1f cells by electroporation at 1500 V using a MicroPulser Electroporator (Bio-Rad, Hercules, CA, USA). Competent cells were obtained by incubation with 0.1 M TE-LiAc for 45 min at 30 °C in the dark and then after adding 2.5 mM Dithiothreitol (DTT) over the next 15 min. Cells were washed with 0.1 M sorbitol.

Cells after electroporation were allowed to grow on Petri dishes with YNB (Merck, Germany) supplemented with Dropout Medium Supplement without uracil (Merck, Germany) and 6 mg/mL chloramphenicol, to which *Y. lipolytica* is resistant, at 30 °C in the dark for 60–72 h.

### 2.4. Western Blotting

For protein isolation, pZ-0, pZ-eGFP, and pZ-HBx-eGFP mutant cells harvested at the exponential growth phase (OD = 0.6) were incubated for 30 min at 29 °C with 10 U/mL zymolyase, pelleted (5600 g, for 5 min), resuspended in TNE lysis buffer (50 mM Tris-HCl, 150 mM NaCl, 5 mM EDTA, pH 7.5) supplemented with 0.1 M Na_3_VO_4_ (an inhibitor of protases), frozen in liquid nitrogen, and disrupted with 0.4 mm glass beads. Then the undisrupted cells and debris were pelleted (500 g, for 5 min). The supernatant (200 μL) containing the isolated proteins was diluted with 200 μL of 2× SDS-gel loading buffer (50 mM Tris-HCl, 10 mM dithiothreitol, 2% SDS, 10% glycerol, 0.1% Coomassie G-250, pH 6.8) and boiled for 5 min.

Western blotting was performed as follows: proteins were run in 15% SDS–polyacrylamide gel with PageRuler™ Prestained Protein Ladder (each well contained 5 µg of total protein); the proteins were transferred to the PVDF membrane. The detection of eGFP and eGFP-fusions were conducted with primary rabbit polyclonal Anti-GFP antibody and secondary Anti-Rabbit IgG Peroxidase antibody. The immunoblot was developed with SuperSignal™ West Dura Extended Duration Substrate in a ChemiDoc Touch Imaging system (BioRad, Hercules, CA, USA).

### 2.5. Mitochondria Visualization in Cells by SIM Microscopy

For mitochondria staining, *Y. lipolytica* cells were loaded with 500 nM MitoTracker Red CmxRos for 30 min. Stained cells were fixed with 2.5% PFA for 10 min, then washed with 50 mM PBS, pH 5.5, and embedded in mounting medium containing 24% glycerol, 9.6% Mowiol, and 2.5% DABCO in 0.1 M Tris, pH 8.5. Imaging was performed using an inverted motorized microscope Eclipse N-SIM with a PerfectFocus autofocusing system (Nikon). The microscopy system was equipped with a 100× Apo TIRF Oil objective (NA1.49), 488 and 561 nm diode laser, and cooled EM-CCD camera iXonDU-897E (Andor, Belfast, Northern Ireland, UK) under the control of NIS-Elements v. 5.11 (Nikon, Tokyo, Japan) software. Image acquisition, SIM image reconstruction, and data alignment were performed using NIS-Elements (Nikon). Then 3D reconstruction of x, y, and z SIM datasets (z-stacks) was performed using ICY software v.2.5 (Biological Image Analysis Unit, Institut Pasteur, Paris, France). 

### 2.6. Oxidative Stress and Cell Death of Y. lipolytica Cells

Production of intracellular reactive oxygen was determined with Dihydroethidium. Yeast cell viability was detected with Sytox Green Dead Cell Stain [69]. To mitigate oxidative stress induced by HBx expression, the cells were preincubated for 1 h with 250 nM SkQThy, a mitochondria-targeted (transported primarily, if not exclusively, in mitochondria) very efficient antioxidant, consisting of lipophilic cation triphenylphosphonium bonded by a C 10 aliphatic chain with an antioxidant thymoquinone (Thy) having versatile healing abilities [62]. Stained cells were analyzed by flow cytometry with a CytoFlex S flow cytometer (Beckman Coulter, Brea, CA, USA). The data obtained for 20,000 cells were stored and analyzed on a logarithmic scale using CytExpert software v2.4 (Beckman Coulter, USA).

### 2.7. Isolation of Y. lipolytica Mitochondria

Mitochondria were isolated by the method developed in our laboratory [64]. The quality of isolated mitochondrial preparations was judged by respiratory control ratios (RC), ratios of respiratory rate of mitochondria in state 3 respiration induced by the addition of ADP to respiratory rate in state 4 respiration after phosphorylation of the ADP added, as recommended in [70]. 

### 2.8. Monitoring of Oxygen Consumption by Yeast Cells and Mitochondria

Oxygen consumption by yeast cells and mitochondria was monitored amperometrically using a closed Clark-type oxygen electrode in a continuously stirred, thermostatically controlled 1 mL cell. The incubation medium for cells contained 50 mM Tris-phosphate buffer supplemented with 1.3% succinate, pH 5.5. The basic incubation medium for mitochondria contained 0.6 M mannitol, 2 mM Tris-phosphate buffer, pH 7.2, 1 mM EDTA, 20 mM Tris-pyruvate + 5 mM Tris-malate (P + M) or 20 mM succinate. Respiratory rates were expressed as ng-atoms O/min/mg protein. For calculating respiratory rates of cells or mitochondria, it is necessary to know the amount of dissolved oxygen in the cell (cell volume and table data on oxygen solubility at experiment temperature) and the amount of added cells (using flow cytometry) or mitochondria (the volume of mitochondria multiplied by their concentration). Respiratory rates of cells and mitochondria were expressed as ng-atoms O/min/cell number or ng-atoms O/min/mg protein, respectively.

### 2.9. Assessment of the Mitochondrial Membrane Potential 

The potential generated at the inner mitochondrial membrane was assessed with safranine O as a potential-sensitive probe with a DU-650 spectrophotometer (Beckman Coulter, USA) using a two-wavelength mode (511–533 nm) [71]. The basic incubation medium was supplemented with 20 µM safranin O and mitochondria (0.5 mg protein/mL).

### 2.10. Monitoring of Mitochondrial Swelling

Mitochondrial swelling was monitored spectrophotometrically with a Cary 300 Bio spectrophotometer (Varian, Palo Alto, CA, USA) by recording changes in apparent absorbance at 540 nm. The basic incubation media were supplemented with 40 mM KCl and mitochondria (0.5 mg protein/mL).

### 2.11. Assay of ATP Synthesis by Mitochondria

ATP synthesis by mitochondria was recorded by two independent methods, the first one based on a small pH shift during ADP conversion to ATP. ATP synthesis was assayed spectrophotometrically with a DU-650 spectrophotometer (Beckman Coulter, USA) at 557/618 nm using Phenol Red (a pH-dependent dye), as described in [72] with minor modifications [67]. All incubation media and addition solutions were carefully adjusted to pH 7.1. The basic incubation medium was supplemented with 6 μM Ap5A (an inhibitor of adenylate kinase), 5 µM Phenol Red, and mitochondria (0.2 mg protein/mL). The second method was based on the coupling of ATP synthesis with NADP reduction in enzymatic reactions containing NADP, hexokinase, and glucose-6-phosphate dehydrogenase. The basic incubation medium was supplemented with 6 μM Ap5A, 1 mM glucose, 1 mM NADP, hexokinase (10 U/mL), glucose-6-phosphate dehydrogenase (3 U/mL), and mitochondria (0.2 mg protein/mL). ATP synthesis initiated by the addition of ADP was monitored spectrophotometrically with a Cary 300 Bio spectrophotometer (Varian, USA) at 340 nm.

### 2.12. Assessment of Hydrogen Peroxide Production by Mitochondria

The production of hydrogen peroxide by mitochondria was determined fluorometrically by measuring the oxidation of Amplex Red to resorufin coupled with the enzymatic reduction of hydrogen peroxide by horseradish peroxidase. The fluorescence of resorufin was measured at room temperature using a RF 5301 PC spectrofluorophotometer (Shimadzu, Kyoto, Japan) at 563/587 nm excitation and emission wavelengths, respectively. Signals were calibrated with 0–5 μM H_2_O_2_ (determined by absorbance at 240 nm) [67]. The basic incubation medium was supplemented with 0.5 mM EGTA, 20 mM Tris-succinate, 5 µM Amplex Red, horseradish peroxidase (9 U/mL), 6 mM aminotriazole (an inhibitor of catalase), and mitochondria (0.2 mg protein/mL).

### 2.13. Mitochondrial Protein Assay

Mitochondrial protein was determined using the Bradford method [73] with BSA as the standard.

### 2.14. Statistical Analysis

Unless otherwise specified, all experiments with yeast mitochondria were performed at least three times with consistent results. For analysis of mitochondrial morphology, at least fifty cells were examined in each trial. Statistical analyses were performed using the one-way ANOVA test with the post hoc Tukey HSD test. Data were presented as mean ± S.E. from at least three independent replicates.

## 3. Results

### 3.1. Creation and Primary Characterization of Yeast HBx-Expressing Cells

One of the main scientific tasks of the research was to create a simple and adequate model of HCC based on *Y. lipolytica* yeast to study bioenergetic aspects of the pathogenesis of this disease. The yeast *Y. lipolytica* is well suited for this kind of research (see Introduction). In addition, the Po1f strain used in the work, while retaining all the advantages of the yeast *Y. lipolytica*, in contrast to the wild strain, is auxotrophic for uracil and leucine, and has a deletion of the Xpr2 gene encoding an extracellular protease, thus making it possible to use as a selective growth media. 

The genetically modified yeast *Y. lipolytica* expressing HBx and HBx-eGFP was created for the first time (see Materials and Methods). The control *Y. lipolytica* Po1f pZ-0 strain, not having the target proteins but carrying the pZexpress++ integrative plasmid containing the URA3 gene as a prototrophic factor, was also constructed.

Analysis of an electrophoregram showed the presence in the constructed Polf pZ-HBx-eGFP mutant of the target protein fused with eGFP with a molecular mass of 44 kDa, which corresponded to its calculated molecular weight (Figure 1).

As a first step, we examined what changes took place in cells expressing heterologous proteins and focused primarily on respiration rates, induction of oxidative stress, and cell death, and the morphology and dynamics of mitochondria within cells. Expression of heterologous proteins did not affect the growth rate of yeast cells (not shown). 

### 3.2. Morphology of Mitochondria in Yeast Cells

*Y*. *lipolytica* cells are quite small (5–10 µm long); therefore, for mitochondria visualization in cells, we applied structured illumination microscopy with improved resolution. Cells of the control strain (pZ-0) contained numerous mitochondria forming the mitochondrial reticulum (Figure 2, left panel). 

The eGFP expression did not change the mitochondrial morphology (Figure 2, left panel), and eGFP was diffusely distributed in the cytosol. In contrast, in pZ-HBx-eGFP cells, the mitochondrial reticulum was disturbed, and mitochondria were fragmented (Figure 2, left panel), strictly indicating the influence of HBx on the mitochondrial structure. In cells expressing HBx, incubation with 250 nM SkQThy partially restored the mitochondrial reticulum (Figure 2, right panel).

An important conclusion can be drawn from this part. To our knowledge, an adequate yeast model (Po1f pZ-HBx) was developed for the first time to trace HBx-mediated changes in the morphology and structure of mitochondria within the cell. 

### 3.3. Oxidative Stress and Cell Death in Yeast Cells 

ROS generation and cell death in *Y. lipolytica* mutants were determined by using flow cytometry in double stained cells with dihydroethidium (DHE), a marker for oxidative stress in the cell, and Sytox Green, a dye that stains only dead cells. Three cell populations were identified according to the level of dye fluorescence (Figure 3A). The cell population with a low level of fluorescence of both dyes was taken as normal for living cells not subjected to oxidative stress (Figure 3A,B, marked by green). The cell population with high DHE fluorescence and low Sytox Green fluorescence corresponded to living cells experiencing oxidative stress (Figure 3A,B, marked by red). The population with a high level of fluorescence of both dyes corresponded to dead cells (Figure 3A,B, marked by blue). The ratio of the number of cells in a given population to the total number of analyzed cells allowed us to judge the prevailing redox status and viability of cells (Figure 3). In Figure 3, Panel B, the same data are presented in the form of histograms.

It was found that cells expressing HBx experienced slightly stronger oxidative stress than the control strain (Figure 3). 

To mitigate oxidative stress induced by HBx expression, we used the mitochondria-targeted antioxidant SkQThy [62]. Preincubation of pZ-HBx cells with 250 nM SkQThy for 1 h marginally reduced the number of cells with oxidative stress in the total cell population, thereby demonstrating that oxidative stress was caused, at least partially, by mitochondrial reactive oxygen species (mitochondrial ROS, mROS). 

### 3.4. Energy Parameters of Mitochondria Isolated from Yeast Cells

Because there was a mismatch between a slight increase in oxidative stress and clear mitochondrial fragmentation seen in HBx-expressed cells, our next step was to examine the bioenergetics of these cells at the mitochondrial level. Previously we have shown that mitochondrial fragmentation is induced not only by oxidative stress, but also in the presence of a uncoupler [67]. Mitochondria were isolated by differential centrifugation according to the method developed in our laboratory [64]. Since the isolation of mitochondria from yeast is a time-consuming process requiring a lot of material, we compared mitochondria isolated from the Po1f and HBx-expressing cells. The mitochondria isolated from the control Po1f strain fully met the criteria for physiological integrity, they were tightly coupled, and did not differ from mitochondrial preparations usually obtained for this type of yeast [64] (Figure 4A). In contrast, the mitochondria from pZ-HBx cells were partially uncoupled, with the respiratory control ratios two times lower compared to the mitochondria from the control cells (Figure 4B). 

Mitochondria from the two strains respiring on NAD-dependent substrates had almost similar respiratory rates in state 4 respiration, while they significantly differed in state 3 respiration and the uncoupling state (in the presence of the classical uncoupler carbonylcyanide *m*-chlorophenylhydrazone CCCP), being much lower in mitochondrial preparations from pZ-HBx cells (Figure 5). These data were in good agreement with the lower respiratory control values in mitochondria from HBx-expressing cells as compared to the control variant. Although mitochondria from HBx-expressing cells were loosely coupled and had a reduced respiratory rate, they retained unchanged the structure of the respiratory chain, as inferred from the almost complete block of mitochondrial respiration by low concentrations of rotenone (Figure 6A) or antimycin A (Figure 6B) specific inhibitors of complex I and III, respectively. 

In good agreement with the above data, mitochondria from the HBx-expressing mutant were much more sensitive to the action of a depolarizing agent (palmitate) than mitochondria of the control variant (Figure 7). 

Swelling of mitochondria is an energy-dependent process. Amplitude of swelling (normalized on protein content) of mitochondria from HBx-expressing cells was approximately two times lower compared to those from the control variant (Figure 8). The poroformer alamethicin, forming pores of 1 nm in diameter in the membrane and causing maximal swelling of mitochondria [74], was taken as a negative control.

The rate of ATP synthesis was measured by two independent methods (see Methods), with similar results. The rate of ATP synthesis in mitochondria isolated from the pZ-HBx mutant was significantly lower than in the control strain (Figure 9A,B). 

Finally, the rate of production of hydrogen peroxide by isolated mitochondria of the control and HBx-expressing yeasts was measured. Hydrogen peroxide production was measured fluorimetrically by changing the fluorescence level of resorufin (see Materials and Methods). Mitochondrial catalase activity was inhibited by adding aminotriazole. Although the main type of ROS generated by mitochondria is superoxide anion-radical, it is rather quickly converted to hydrogen peroxide. Only an insignificant increase in the production of hydrogen peroxide by the mitochondria of HBx-expressing cells was observed compared to the control cells (Figure 10). These seemingly unexpected results can be easily explained by partial uncoupling of these mitochondria, as inferred from the results presented in Figure 4, Figure 5, and Figure 9. It is known that a decrease of the membrane potential by only 10% reduces ROS (reactive oxygen species) production by 90% [75]. *t*-BHP, a well-known model prooxidant forming ROS as a result of an enhanced lipid peroxidation reaction [76], multiplied the production of hydrogen peroxide by mitochondria, with mitochondria from HBx-expressing cells being more susceptible to oxidative stress than control mitochondria.

Thus, the comprehensive study of energy parameters of mitochondria isolated from HBx-expressing yeast showed that HBx induced mitochondrial dysfunction, primarily a lower degree of coupling of respiration to phosphorylation and a reduced rate of respiration and ATP production, and higher sensitivity to depolarizing agents and oxidative stress.

## 4. Discussion

Collectively, data from Western blotting (Figure 1) and SIM fluorescent microscopy (Figure 2) show that to the best of our knowledge, for the first time, genetically modified *Y. lipolytica* yeast expressing HBx and HBx-eGFP proteins was created. Therefore, we fulfilled the main task that we set. Moreover, the resulting constructs have turned out to be useful for deciphering the effects of HBx expression on the morphology and functions of mitochondria.

The control experiments showed that expression of eGFP did not affect the morphology of mitochondria, it was diffusely distributed in the cytoplasm, and it did not contribute to the formation of protein aggregates. In contrast, in pZ-HBx-eGFP-expressing yeast, protein aggregates were clearly seen. Moreover, they were localized at the cell periphery, where mitochondria are predominantly located, and mitochondria were largely fragmented (Figure 2). Importantly, incubation with 250 nM mitochondria-targeted antioxidant SkQThy partially restored the mitochondrial reticulum and prevented the formation of protein aggregates (Figure 2). 

In eukaryotic cells, high energy demand requires mitochondria to constantly fuse, divide, and move along the cytoskeleton. These processes are mainly provided by dynamine-like proteins regulated by various protein–protein interactions and post-translational modifications [77]. Mitochondrial dynamics (fusion and fragmentation) are thought to play an important role in the maintenance of mitochondrial health and are part of the mitochondrial quality and quantity control mechanism [78,79,80]. 

Three main conclusions can be drawn from these results. Firstly, expression of HBx protein had prominent effects on mitochondria morphology. Secondly, these adverse effects could be mitigated by mitochondria-addressed antioxidants, which means that to some extent these effects could be induced by the mitochondrial ROS (mROS). Thirdly, as we obtained data coinciding with the literature data from other models (see Introduction), we really believe that we have managed to obtain an adequate yeast model for the study of many biological processes related to the influence of HBx. 

Cells expressing HBx experienced slightly stronger oxidative stress than the control strain (Figure 3), and preincubation with SkQThy reduced the number of cells with oxidative stress and dead cells in the total cell population, thereby reinforcing the notion that at least partially, oxidative stress was caused by mitochondrial reactive oxygen species (mitochondrial ROS, mROS) [76]. There is a consensus that mROS, primarily superoxide anion radical, is the starting form of ROS to propagate (see [81,82]). Previously we showed that SkQThy alleviated and even prevented oxidative stress induced by prooxidants [62]. Basically, lipophilic mitochondria-addressed antioxidants have advantages over conventional water-soluble antioxidants; their concentrations in mitochondria are increased several orders of magnitude in comparison with the initial low nontoxic concentrations [83]. In addition, having in their composition a natural component of the electron-transporting chain, they can be effectively recharged. 

Recently, using time-lapse microscopy and fluorescent dyes, electively visualizing different cell compartments, we found that the *t*-BHP-induced oxidative stress initially developed only in mitochondria, starting almost immediately after contact with the prooxidant and then was progressively increased. ROS production in mitochondria far preceded the appearance of the generalized oxidative stress detected in the cytoplasm in the volume of the entire cell. Once started, the generalized oxidative stress was increased, but only after a prolonged lag-period. A significant decrease in the production of mitochondrial ROS was observed during preincubation of yeast cells with a mitochondria-targeted antioxidant. Importantly, mitochondrial fragmentation preceded the development of generalized oxidative stress and could be induced by much lower concentrations of prooxidant as compared to generalized oxidative stress and cell death [76]. Mitochondrial dysfunction contributes to cell death at the early stages of the development of various diseases, and mitochondrial fragmentation is one of the earliest symptoms [84,85,86,87].

Analysis of isolated mitochondria showed that HBx expression had a significant influence on the bioenergetics functions of mitochondria, as inferred from the decreased respiratory rate (Figure 4), while maintaining the overall structure of the respiratory chain (Figure 6), higher sensitivity to the depolarizing agent (Figure 5), and reduced ATP formation (Figure 9) and respiratory control values (characterizing coupling degree of respiration and phosphorylation), as compared to the control mitochondria (Figure 4). We believe that mitochondrial dysfunction in the HBx-expressing cells was the main cause of their fragmentation. We showed this possibility earlier [67].

In sum, we offered the first highly promising yeast model for studying the impact of HBx on the bioenergetics, redox-state, and dynamics of mitochondria in the cell, and the cross-talk between these parameters. This fairly simple model can be used as a platform for the rapid screening of compounds, which can mitigate the harmful effects of HBx. Based on the data obtained in the work, we could recommend a family of mitochondrial-directed mitochondrial lipophilic antioxidants that bind to active oxygen forms in mitochondria (at the places of their formation) and prevent mitochondrial fragmentation, the earliest manifestation of many pathologies. It would be wise to take these compounds purposefully to prevent these pathologies.

## Figures and Tables

**Figure 1 microorganisms-10-01817-f001:**
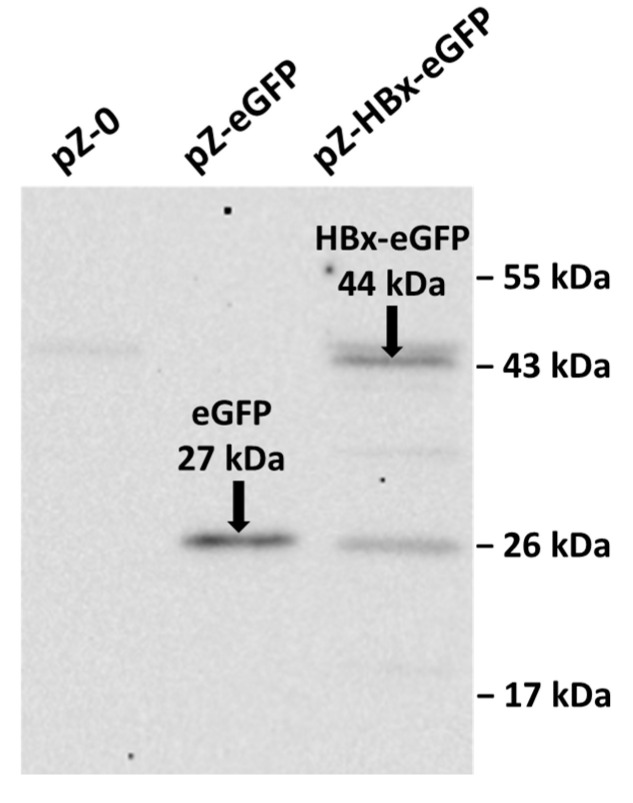
Western blot analysis of eGFP fusion proteins from pZ-0, pZ-eGFP, and pZ-HBx-eGFP strains of *Y. lipolytica*. The eGFP and HBx-eGFP signals are indicated by arrows. On the right, the molecular weights of the markers are shown.

**Figure 2 microorganisms-10-01817-f002:**
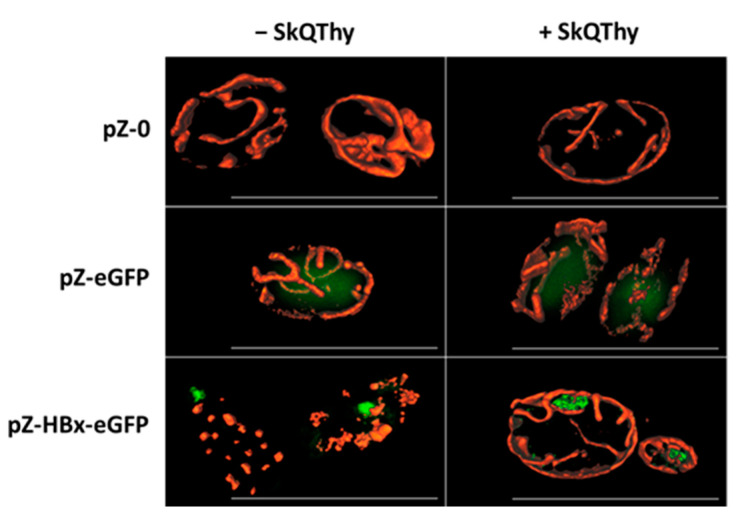
Mitochondrial morphology of *Y. lipolytica* mutants. Effect of SkQThy. eGFP fluorescence shown in green. Bars are 10 µm. Cells were preincubated without SkQThy (**left** panel) or with 250 nM SkQThy (**right** panel) for 1 h, then washed with 50 mM PBS, pH 5.5, and stained with 500 nM MitoTracker Red CmxRos (shown in red) for 30 min.

**Figure 3 microorganisms-10-01817-f003:**
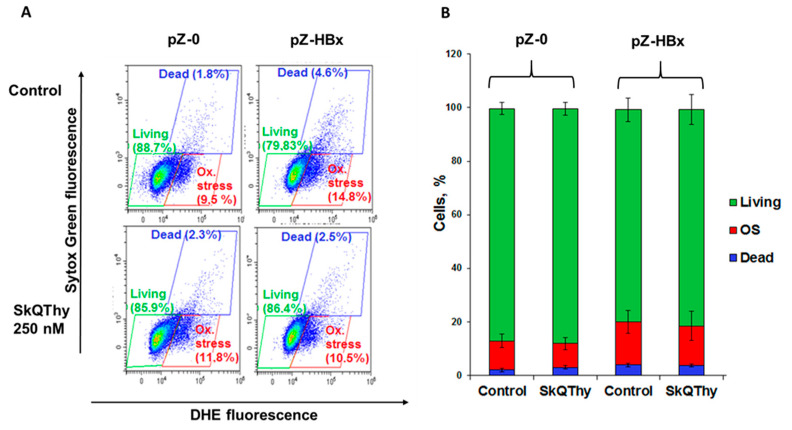
Oxidative stress and cell death in *Y. lipolytica* strains. Panel (**A**)—flow cytometry measurement; Panel (**B**)—the same results are presented in the form of histograms summarizing results from three independent experiments.

**Figure 4 microorganisms-10-01817-f004:**
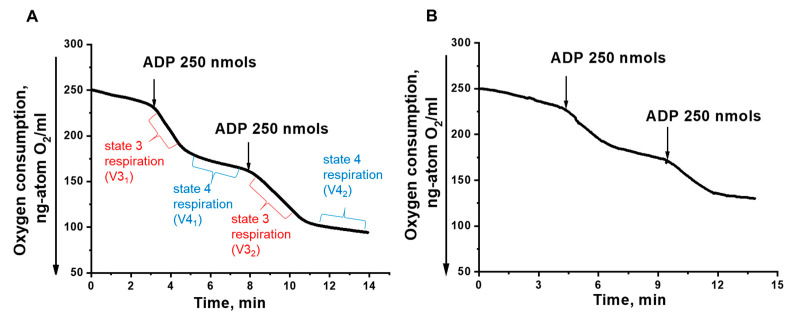
Amperometric recording of oxygen consumption by *Y. lipolytica* mitochondria. (**A**) *Y. lipolytica* Po1f; (**B**) *Y. lipolytica* Po1f pZ-HBx. The basic incubation medium contained 0.6 M mannitol, 2 mM Tris-phosphate, pH 7.2, 1 mM EDTA, 20 mM Tris-pyruvate + 5 mM Tris-malate and mitochondria (0.5 mg protein/mL). Respiratory control ratios upon successive ADP additives were: (**A**) 4.3, 5.2; (**B**) 2.3, 2.2.

**Figure 5 microorganisms-10-01817-f005:**
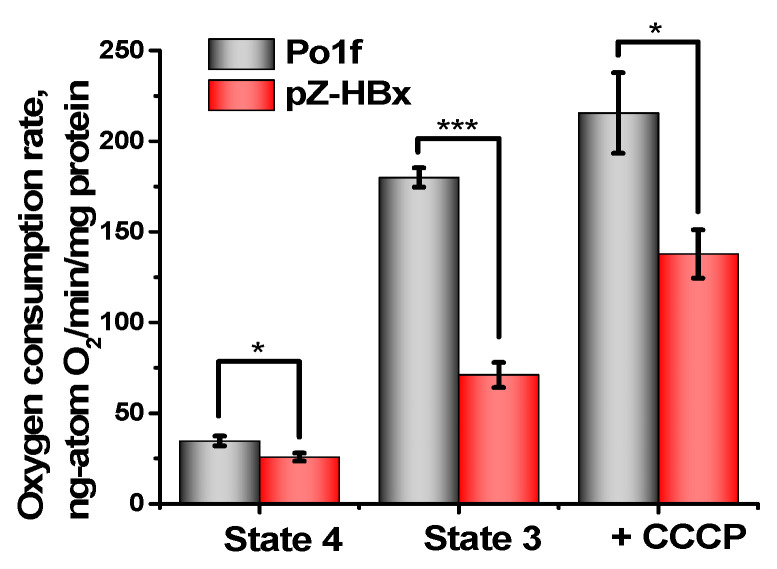
Oxygen consumption rates by *Y. lipolytica* mitochondria in states 3 and 4, and uncoupled states. When indicated, 2 µM CCCP was added. The statistical analyses were carried out by the one-way ANOVA test. ***: *p* < 0.001; *: 0.01 < *p* < 0.05.

**Figure 6 microorganisms-10-01817-f006:**
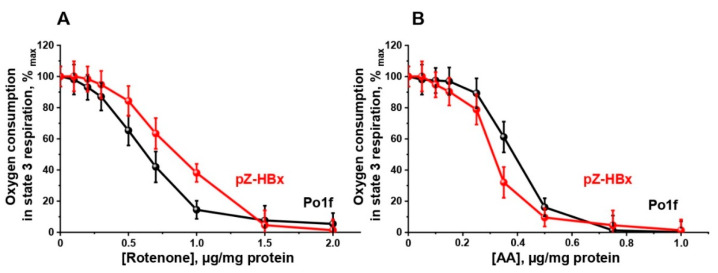
Amperometric recording of oxygen consumption by *Y*. *lipolytica* mitochondria in state 3 respiration as affected by respiratory inhibitors: rotenone (**A**) and antimycin (**B**). Incubation medium was supplemented with 1 mM ADP and mitochondria (0.5 mg protein/mL).

**Figure 7 microorganisms-10-01817-f007:**
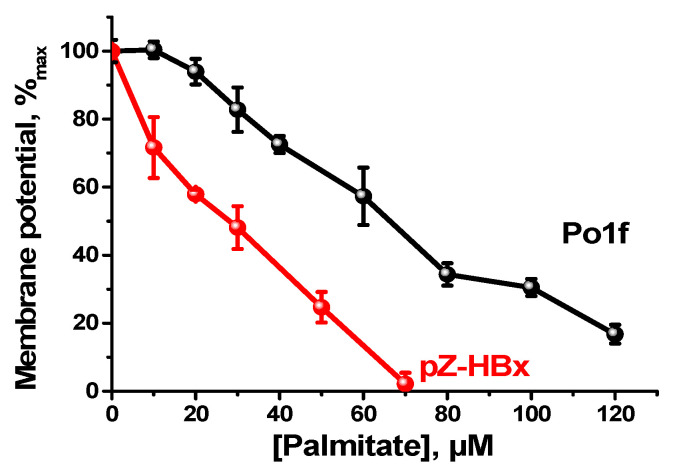
Membrane depolarization of *Y. lipolytica* mitochondria by palmitate. The basic incubation medium was supplemented with 20 mM safranine O and mitochondria (0.5 mg protein/mL).

**Figure 8 microorganisms-10-01817-f008:**
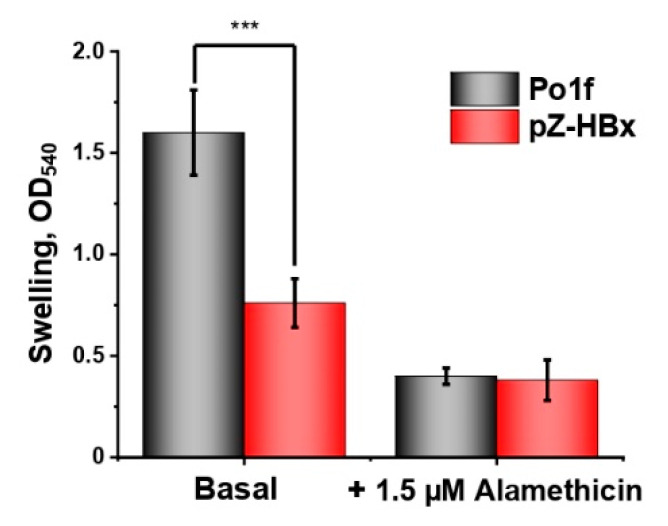
Swelling of *Y. lipolytica* mitochondria. The basic incubation medium was supplemented with 40 mM KCl and mitochondria (0.2 mg protein/mL). The statistical analyses were carried out by the one-way ANOVA test. ***: *p* < 0.001.

**Figure 9 microorganisms-10-01817-f009:**
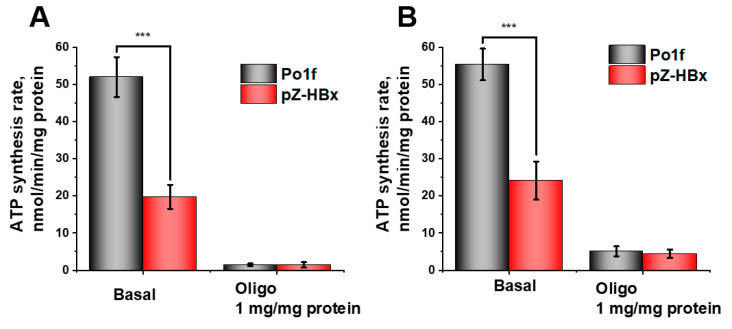
ATP production by *Y. lipolytica* mitochondria. (**A**) The basic incubation medium was supplemented with 5 µM Phenol Red, 6 μM Ap5A (an inhibitor of adenylate kinase), and mitochondria (0.2 mg protein/mL); (**B**) The basic incubation medium was supplemented with 6 μM Ap5A, 1 mM glucose, 1 mM NADP, hexokinase (10 U/mL), glucose-6-phosphate dehydrogenase (3 U/mL), and mitochondria (0.2 mg protein/mL). The statistical analyses were carried out by the one-way ANOVA test. ***: *p* < 0.001.

**Figure 10 microorganisms-10-01817-f010:**
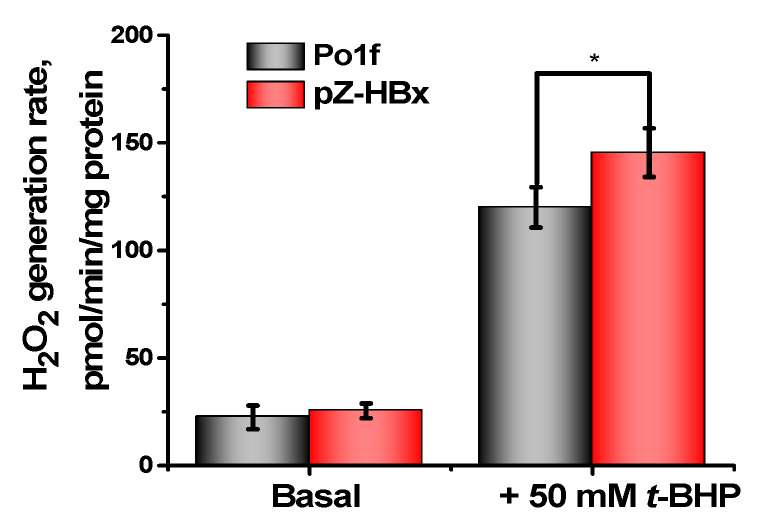
H_2_O_2_ generation by *Y. lipolytica* mitochondria. The statistical analyses were carried out by the one-way ANOVA test. *: 0.01 < *p* < 0.05.

**Table 1 microorganisms-10-01817-t001:** List of strains.

Strain	Description
Po1f	MatA, leu2-270, ura3-302, xpr2-322, axp-2
pZ-0	Po1f + pZ URA3, xpr2Δ
pZ-eGFP	Po1f + pZ-eGFP URA3 xpr2Δ
pZ-HBx	Po1f + pZ-HBx URA3 xpr2Δ
pZ-HBx-eGFP	Po1f + pZ-HBx-eGFP URA3 xpr2Δ

**Table 2 microorganisms-10-01817-t002:** List of primers.

Primer	Sequence
HBx-BbsI-Fw1	TAGAAGACGCAATGGCTGCTAGGCTGTGC
HBx-BbsI-Rev1	TAGAAGACGCGCGCTTAGGCAGAGGTGAAAAAGTTGC
HBx-BbsI-rev2	TAGAAGACGCGGCAGAGGTGAAAAAGTTGC
eGFP-BbsI-Fw1	TAGAAGACTAAATGGTGAGCAAGGGCGAGGAG
eGFP-BbsI-Rev1	TAGAAGACGCGCGCTTACTTGTACAGCTCGTCCATG
eGFP-BbsI-Fw2	TAGAAGACATTGCCATGGTGAGCAAGGGCGAG

## Data Availability

The data used to support the findings of this study are available from the corresponding author upon request.

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
