# Peer review of "The First Yarrowia lipolytica Yeast Models Expressing Hepatitis B Virus X Protein: Changes in Mitochondrial Morphology and Functions"

_microorganisms, 2022, doi:10.3390/microorganisms10091817_

Round 1

Reviewer 1 Report

The manuscript: “The first Y. lipolytica yeast models expressing HBx and HBx120 

The authors describe: The generation of a genetically modified yeast Y. lipolytica, expressing a hepatitis B viral protein, HBx; and the carboxyl terminal fragment, HBx120. This new model will facilitate the study of roles of HBx on bioenergy, oxidative stress and mitochondrial dynamics. The topic and results are highly relevant and majority of the experiments showed good quality and include the proper controls. An extensive introduction and discussion was included. However this work will need several modifications and clean up writing.

 Concerns and suggestions are listed below.

 1.    Strongly suggest to improve writing/editing, especially the results section and figure legends.

2. I suggest to modify the title in order to give more information about this paper.  

3.    In order to improve the quality of the results section, authors should indicate aim of each approach. In other words, mention what is the question to be answer and how the authors resolved it. Finally, the results are poorly written, you may improve them and be more explicit with the interpretations of the results.

4.    Authors must improve the figure legends. Figure legend must give more information with less words. For example, in figure 5 and 6, remove the constitution of the medium, since it is already described on the methods section.

5.    Reorder materials and methods. It will facilitate the lecture of the paper if materials and methods were order in similar way as they are mention in result sections. In addition, separate section 2.4 as single section as well as oxygen consumption, potential generated at the inner mitochondrial membrane, mitochondrial swelling, ATP synthesis and production of hydrogen peroxide.

6. Demonstrate the expression of the different HBx and HBx120 proteins by western blot or immunofluorescence.

7. If possible, mix figure 1 and figure 2. And indicate the difference between them.

8. Figure 2, combine the three graphs in order to get one single graph.

9. Provide enough information of the chemicals and constructions. For example: Add the description “mitochondrial-target antioxidant” to SkQThy in the methods (section 2.5., line 247); figure legend 2 (line 304). The description “prooxidant” to t-BHP in in figure legend 2 (line 306). And a brief description of HBx120 in the abstract.

10. Line 266 and 267 should be move to 2.1. Chemical reagents.

11. Provide a list of abbreviations or include them in the text. Some abbreviations are missing (wtHBx, trHBx line 102 and 103; CCCP line 374 and figure legend 5).

12. Remove typos like: SKThy (line 26); “s” line 86.

Author Response

Thank you for your kind attitude and valuable advice. We fully agree with all comments.  Accordingly, the revised version was substantially corrected: we modified the title; supplemented the text with the latest data; extended the Discussion section; reformatted the whole text, especially the Materials and Methods and Results sections, answered all questions, edited the text.

 First of all, we would like to thank the referee for useful comments and advises and are ready to answer the questions raised.

  1. “Strongly suggest to improve writing/editing, especially the results section and figure legends”

Response

We have consulted a colleague who is considered as a good English speaker. In short, we have done our best. 

  1. “I suggest to modify the title in order to give more information about this paper|”.

Response

The title was modified

  1. “ In order to improve the quality of the results section, authors should indicate aim of each approach. In other words, mention what is the question to be answer and how the authors resolved it. Finally, the results are poorly written, you may improve them and be more explicit with the interpretations of the results”.

Response

We have made an attempt to do this in the corrected version

  1. “Authors must improve the figure legends. Figure legend must give more information with less words. For example, in figure 5 and 6, remove the constitution of the medium, since it is already described on the methods section”.

Response

We have rewritten Figure legends, trying not to wake up the information presented in the section Materials and Methods

  1. “Reorder materials and methods. It will facilitate the lecture of the paper if materials and methods were order in similar way as they are mention in result sections. In addition, separate section 2.4 as single section as well as oxygen consumption, potential generated at the inner mitochondrial membrane, mitochondrial swelling, ATP synthesis and production of hydrogen peroxide”

Response

We have rewritten the Materials and Methods section as recommended by the reviewer.  Results description aligned with the new version of this section

  1. “Demonstrate the expression of the different HBx and HBx120 proteins by western blot or immunofluorescence”.

Response

We understood from the very beginning that the final proof that HBx is expressed in yeast cells would be the use of specific antibodies. It turned out, however, that isolation of HBx and therefore getting antibodies to it was a difficult task. All attempts of our colleagues, experts in protein science, to obtain HBx in a pure form were unsuccessful. They were joking bitterly that the protein formed aggregates “even from looking at it”.  So we had to use a different approach - to fuse HBx and GFP and to follow localization of chimeric protein within the cell using fluorescent microscopy. We realize that for final proof this may not be enough, but perhaps enough, for the first step.

  1. If possible, mix figure 1 and figure 2, and indicate the difference between them. 8. Figure 2, combine the three graphs in order to get one single graph.

Response

We combined figures 1 and 2, removing the results with t-BHP, explaining in the Figure legend that in the panel B the results shown in the panel A are in the form of histograms

  1. “Provide enough information of the chemicals and constructions. For example: Add the description “mitochondrial-target antioxidant” to SkQThy in the methods (section 2.5., line 247); figure legend 2 (line 304). The description “prooxidant” to t-BHP in in figure legend 2 (line 306). And a brief description of HBx120 in the abstract”.

Response

It was done.

  1. 10. “Line 266 and 267 should be move to 2.1. Chemical reagents”.

Response

It was done.

  1. Provide a list of abbreviations or include them in the text. Some abbreviations are missing (wtHBx, trHBx line 102 and 103; CCCP line 374 and figure legend 5).

Response

Abbreviations have been deciphered in the text

  1. Remove typos like: SKThy (line 26); “s” line 86.

Response

It was done.

Reviewer 2 Report

The paper of Epremyan et al et al describes  the expression of HBV protein HBx in the yeast  Yarrowia lipolytica  as a model to study the impact of HBx protein on energy metabolism and mitochondrial activities. The paper is interesting but should be improved since some experimental methods need a more detailed description and some necessary controls are lacking.
What is the vector used pZexpress++ ?? (I was not able to retrieve any information on it..) ( no reference was given , it is a bacterial plasmid? is an integrative yeast vector?, what is the promoter used for the expression of the cloned genes??). The authors  did not give any evidence that the cloned genes (HBx and HBx120) were correctly expressed in Yarrowia ( at least a Western blot should be done to evaluate the level of expression...). All the paragraph describing the plasmids construction  (line 151-178) should be rewritten in a more comprehensible way. For example what is the sequence of HBx120 ?? it is a  truncated protein with the aa 1-120 or  aa 120-154??
The authors used two different eGFP fusions, ( eGFP-HBx  and Hbx-eGFP) why? The presence of eGFP  in N or C terminus could change the correct intracellular localization of the fusion protein...It has been verified?
The authors claimed that the  cells expressing of HBx  are more sensitive to oxidative stress ( line 310-311) but this is not evident in the Fig.1 and 2 where the fraction of Ros producing cells and of dead cells  in presence of t-BHP were similar to the control!!!
The discussion should be rewritten is too long and contains repetition of experimental results. All the paragraphs describing the effects of HBx protein in mammalian cells (line 460 -490) should be moved to the Introduction section...and the lines 507 to 516 are a repetition of methods used..
Minor questions:
Title: is not clear   Y should be changed in Yarrowia and HBx and HBx120 needs some more informative description ( like  "  expressing Hepatitis B virus Hbx protein..".) what is HBx120??
In the Abstract, line 26 SkThy   should be   SkQThy
line 28 pZ-Hbx cells ?? ( may be in cells expression Hbx protein..)
Line 77:    trans factor activity ??   may be transcriptional activation activity
Line 93 regulation of HBx replication ??  may be  regulation of HBV replication
Lines 131 and 134  restrictase??  restriction endonuclease
Line 138 E coli XL1-Blue ( give a reference, obtained from?)
Line 242  " production of intracellular reactive oxygen with Sytox Green" ??? what means??
Sytox Green stains dead cells...

Results:
Line 278  expressing HBx120, HBx120-eGFP etc..  what is HBx120??  Where are the evidences for the expression of the cloned proteins??
Lines 310-311   " increased manifold as compared..."  is not true at least for the data in Fig.1 and  Fig.2 !!
Line 319  SkThy    ... SkQThy
Line 317 expectedly??
Line 370 (Fig.4) Respiratory ratios... how were calculated?
Lines 372-374  please explain better  " state 4 respiration and state 3 respiration"...

Author Response

Thank you for your kind attitude and valuable advice. We fully agree with all comments.  Accordingly, the revised version was substantially corrected: we modified the title; supplemented the text with the latest data; extended the Discussion section; reformatted the whole text, especially the Materials and Methods and Results sections, answered all questions, edited the text.

First of all, we would like to thank the referee for useful comments and advises and are ready to answer the questions raised.

“The paper of Epremyan et al et al describes  the expression of HBV protein HBx in the yeast  Yarrowia lipolytica  as a model to study the impact of HBx protein on energy metabolism and mitochondrial activities. The paper is interesting but should be improved since some experimental methods need a more detailed description and some necessary controls are lacking.”

“What is the vector used pZexpress++ ?? (I was not able to retrieve any information on it..) ( no reference was given , it is a bacterial plasmid? is an integrative yeast vector?, what is the promoter used for the expression of the cloned genes??).”

Lines 179-184. Added pZexpress++ vector description

Response (New information from the revised version is marked by green).

To create target genetic constructs, the pZ-express++ plasmid was selected with a hybrid hp4d promoter dependent on the growth phase, and a ZETA transposon sequence with multiple homology in the Y. lipolytica genome was chosen, which during recombination ensures a high copy number of the plasmid and, as a result, a high level of expression of the target protein. The Plasmid also has an ampicillin resistance gene and a prototrophic factor for uracil URA3 from the Y. lipolytica genome.

 “The authors did not give any evidence that the cloned genes (HBx and HBx120) were correctly expressed in Yarrowia (at least a Western blot should be done to evaluate the level of expression...). All the paragraph describing the plasmids construction (line 151-178) should be rewritten in a more comprehensible way. For example what is the sequence of HBx120 ?? it is a truncated protein with the aa 1-120 or aa 120-154??”

Response

We understood from the very beginning that the final proof that HBx is expressed in yeast cells would be the use of antibodies. It turned out, however, that isolating the protein and therefore getting antibodies to it was a difficult task. Our colleagues from another institute have been trying unsuccessfully to achieve this goal for three years as part of a common project. They were joking bitterly that the protein formed aggregates “even “even from looking at it”.  . So we had to use a different approach - to get the chimeric protein by fusing HBx and GFP to see if it was localized inside the cell. We selected the optimal conditions for the activity of the promoter, under which a high fluorescence of eGFP was observed in the cells. Since all the target proteins were under the same promoter, we considered that under these conditions we can expect normal expression of HBx and HBx120 proteins. Indeed, by using structural illumination microscopy with improved resolution, we have been able to show that the chimeric protein forms aggregates within in the cell and these aggregates are localized at the cell periphery, where the mitochondria are predominantly located. We realize that for final proof this may not be enough, but perhaps enough, for the first step. In the paragraph describing plasmids construction we added description of genetic constructs and the vector.

  1. Materials and methods

2.3. Plasmid and yeast strain construction

Primer design was based on the nucleotide sequences of the genes encoding HBx and eGFP so that the sequence of the PCR product consisted of the full-length HBx nucleotide sequence, the sequence encoding the 1-120 N-terminal amino acids of HBx (HBx120), and the full-length eGFP nucleotide sequence.

The abstract was edited

For the first time, genetically modified yeast Y. lipolytica was created, expressing the hepatitis B virus core protein HBx, its C-end truncated form at 120 amino acid - HBx120, as well as variants fused with eGFP at the C-end.

“The authors used two different eGFP fusions, ( eGFP-HBx  and HBx-eGFP) why? The presence of eGFP  in N or C terminus could change the correct intracellular localization of the fusion protein...It has been verified?”

Response

eGFP itself under normal conditions does not aggregate in cells, does not have a special localization, being diffusely distributed in cells. This can be seen in Fig. 1.

We made constructions containing HBx and HBx120 fused with GPF at both C and N termini. Protein expression was judged by the appearance of green fluorescence in transformed cells. However, mutant cells with HBx and HBx120 fused with GPF at the N-terminus did not display noticeable green fluorescence, for unknown reasons (may be conformational complications).  Just the opposite, in mutants with eGFP s fused at the C terminus, due to normal fluorescence, we were able to assess both the aggregation and distribution of the target proteins. Therefore, only these mutants were used in this work. We have removed from the text all references to mutants in which eGFP was fused at the N terminus: from Tables 1 and 2, lines 159-165, line 185 and line 289.

“Title: is not clear   Y should be changed in Yarrowia and HBx and HBx120 needs some more informative description ( like  "  expressing Hepatitis B virus Hbx protein..".) what is HBx120??”

Response

The title was modified:

The first Yarrowia lipolytica yeast models expressing Hepatitis B virus X protein and its C-truncated form HBx120: сhanges in bioenergetic functions and the therapeutic effect of a mito-chondria-targeted antioxidant.

“Line 77:    trans factor activity ??   may be transcriptional activation activity” Хорен

Response

trans factor activity was changed to transcriptional activation activity.

“Line 93 regulation of HBx replication ??  may be  regulation of HBV replication” Хорен

Response

HBx replication was changed to HBV replication.

“Lines 131 and 134  restrictase??  restriction endonuclease” Хорен

Response

Restrictase was changed to restriction endonuclease.

“Line 138 E coli XL1-Blue ( give a reference, obtained from?)”

Response

  1. Materials and methods

2.2. Cell cultures

Escherichia coli strain XL1-Blue (Evrogen, Russian Federation) was used in the work

“Line 278  expressing HBx120, HBx120-eGFP etc..  what is HBx120??  Where are the evidences for the expression of the cloned proteins??”

Response

See above

Round 2

Reviewer 2 Report

The paper of Epremyan et al et al describes  the expression of HBV protein HBx in the yeast  Yarrowia lipolytica  as a model to study the impact of HBx protein on energy metabolism and mitochondrial activities. The revised paper is clearly  improved  but I still have  some questions regarding the results and  the lack of some necessary controls.
The authors used the vector  pZexpress++  but no references are given (is a plasmid constructed by the authors?? or it has been alreay used by other lab. ?  a proper citation is necessary). Has been obtained by ..(what is the source of this vector?).
The authors  did not give any evidence that the cloned genes (HBx and HBx120) were correctly expressed in Yarrowia ( at least a Western blot should be done to evaluate the level of expression...). The authors did not used specific antibodies for Hbx protein, but if these antibodies were not available at least a Western blot with anti-eGFP antibodies should be done!- the anti eGFP antibodies are commercially available!!

The authors claimed that the  cells expressing of HBx  are more sensitive to oxidative stress  but this is not  so evident in the Fig. 2 where the fraction of Ros producing cells increases by a 9.5% of control cells to 14.8% of Hbx expressing cells ( I suppose that this  experiment was performed at least three times, and that one of them was shown as an example in Fig.2A. If so the mean of the three independent experiments should be given...and the data reported in Fig.2B should be relative to all the experiments..i.e. with a mean and SD. This is very important to be sure that the observed increase in oxidative stress is significant..)

Author Response

Thank you very much for your valuable comments and advice to help correct some of the inaccuracies and make the manuscript stronger.

We are ready to answer the questions raised.

Comments and Suggestions for Authors

The revised paper is clearly improved, but I still have some questions regarding the results and the lack of some necessary controls.

“The authors used the vector  pZexpress++  but no references are given (is a plasmid constructed by the authors?? or it has been alreay used by other lab. ?  a proper citation is necessary). Has been obtained by ..(what is the source of this vector?)”.

Response

The pZ-express++ plasmid was kindly provided by Dr. Laptev I.A. from "State Research Institute of Genetics and Selection of Industrial Microorganisms of the National Research Center" Kurchatov Institute", Moscow, Russian Federation, State Research Institute of Genetics and Selection of Industrial Microorganisms of the National Research Center" Kurchatov Institute", Moscow, Russian Federation. In the revised version (Round 2) we indicated this in the section 2.3. Plasmid and yeast strain construction.

“The authors  did not give any evidence that the cloned genes (HBx and HBx120) were correctly expressed in Yarrowia ( at least a Western blot should be done to evaluate the level of expression...). The authors did not used specific antibodies for Hbx protein, but if these antibodies were not available at least a Western blot with anti-eGFP antibodies should be done!- the anti eGFP antibodies are commercially available!!”

Response

In our previous response we wrote “We understood from the very beginning that the final proof that HBx is expressed in yeast cells would be the use of antibodies. It turned out, however, that isolating the protein and therefore getting antibodies to it was a difficult task”. However, it seemed to us that diffusely distributed green fluorescence in the cytosol of the e-GFP cells expressed GFP, in contrast to the discrete prevailing GFP-related green fluorescence in HBX-expressing cells at the periphery of the cell (where mitochondria are located) could be a sufficient argument that the transformation of the cells has been successful, correct. But your arguments are hard to resist. Therefore we have put a special experiment with Western blotting. In the new version of the manuscript, the results are presented in Fig. 1. It can be clearly seen that cells expressing only GFP protein did contain this protein with expected, according to the literature, molecular mass. This could be a direct proof of the whole protocol that we have developed and used for protein cloning. What’s important is that there was also a protein fused, judging by its molecular mass, with HBx. Together, the results, obtained with the help of Western blot and SIM fluorescent microscopy (Fig. 2 in the new version) have convinced us that really we managed to get yeast model with expressed HBx.

Significant changes to the text were made: expanded the list of reagents used, added a description of the method and the results obtained.

But we have sustained losses along the way. Since the time for the resubmitting the manuscript was limited (5 days only, later this period was extended for three days), we were not able to get reproducible data with Western blotting for the cells expressing HBx-120 protein. Therefore we were had to delete all the data obtained with this protein.

“The authors claimed that the cells expressing of HBx are more sensitive to oxidative stress  but this is not  so evident in the Fig. 2 where the fraction of Ros producing cells increases by a 9.5% of control cells to 14.8% of Hbx expressing cells ( I suppose that this  experiment was performed at least three times, and that one of them was shown as an example in Fig.2A. If so the mean of the three independent experiments should be given...and the data reported in Fig.2B should be relative to all the experiments..i.e. with a mean and SD. This is very important to be sure that the observed increase in oxidative stress is significant..)”

Response

We fully agree with your arguments. In the corrected version (Round 2), in Fig. 2, Panel B, the results of three independent experiments, average values and standard deviations are presented We also corrected the text (Results and Discussion sections) by saying that “cells expressing HBx experienced slightly stronger oxidative stress than the control strain”. Moreover, given these and our early results, we’ve come to the conclusion that the main cause of the observed mitochondrial fragmentation in HBX-expressing cells is not the oxidative stress, but rather mitochondrial dysfunction, which we have ample evidence of when studying isolated mitochondria.
